# Risk Reduction Optimization of Process Systems under Cost Constraint Applying Instrumented Safety Measures †

**Aleksandr Moshnikov ***  **and Vladimir Bogatyrev**

Faculty of Software Engineering and Computer Technigue, ITMO University, 197101 Saint-Petersburg, Russia; vladimir.bogatyrev@gmail.com

* Correspondence: moshnikov.alex@gmail.com
† This paper is an extended version of our report: Moshnikov A. "Process safety instrument system optimization by Monte-Carlo method" in the Majorov International Conference on Software Engineering and Computer Systems (MICSECS 2019), Saint-Petersburg, Russia, 12–13 December 2019.

**Abstract:** This article is devoted to an approach to develop a safety system process according to functional safety standards. With the development of technologies and increasing the specific energy stored in the equipment, the issue of safety during operation becomes more urgent. Adequacy of the decisions on safety measures made during the early stages of planning the facilities and processes contributes to avoiding technological incidents and corresponding losses. A risk-based approach to safety system design is proposed. The approach is based on a methodology for determining and assessing risks and then developing the necessary set of safety measures to ensure that the specified safety indicators are achieved. The classification of safety measures is given, and the model of risk reduction based on deterministic analysis of the process is considered. It is shown that the task of changing the composition of safety measures can be represented as the knapsack discrete optimization problem, and the solution is based on the Monte Carlo method. A numerical example is provided to illustrate the approach. The considered example contains a description of failure conditions, an analysis of the types and consequences of failures that could lead to accidents, and a list of safety measures. Solving the optimization problem used real reliability parameters and the cost of equipment. Based on the simulation results, the optimal composition of the safety measures providing cost minimization is given. This research is relevant to engineering departments, who specialize in planning and designing technological solutions.

**Keywords:** risk reduction; safety instrumental systems; discrete optimization; system design; Monte-Carlo method; system reliability

## 1. Introduction

With the development of technologies and increasing the specific energy stored in the equipment, the issue of safety during operation becomes more urgent. To ensure safety, emergency protection systems have been widely used. As examples of industrial systems that fit the description, we can consider a polar crane, a chemical plant reservoir system, and a turbine. At the heart of the development of such protection systems is the international standard IEC 61511 [1], which introduces the term "safety instrument system" (SIS) and defines it as a system consisting of sensors, logic solvers, and final element controls. Together they implement one or more functions that provide safety [2]. Such systems may contain a set of safety features that act as layers or barriers aimed at deeply layered risk reduction as the first level of protection, we can consider a distributed control system [3], which is designed to ensure

the technology of the process and the formation of control in the normal operation of the equipment. The next barrier is the emergency shutdown system (implemented on the SIS), which brings the object to a safe controlled state. The development of the design of the SIS for industrial facilities is associated with the choice of architecture, nomenclature of components, aspects related to the discipline of service and additional measures to guarantee the development [4]. The content of the article is devoted to optimizing the composition of safety measures, that is, the priority is given to the approach of how to protect the technological process in the event of equipment failures. The first part of the article also focuses on how such development should be carried out, i.e., on the organization of the life cycle in terms of development, namely, how the process of security analysis is related to development and how the main stages of development are provided with the help of regulatory documents.

## 2. Risk Reduction Approach

### 2.1. Relationship of the Safety Analysis and the Design Process

Safety properties are set during the design process. This is ensured by applying a special development lifecycle-focused on safety. At the same time, the safety analysis process takes place in parallel with the development of the main documents. As a result of this approach, an array of protective measures is formed, some of which can be transferred from previous successful projects and applications.

A risk-based approach is used to ensure safety requirements, which consists of close integration of equipment development and safety analysis processes. Below is a detailed description of the basic safety analysis steps during design.

1.  Safety lifecycle planning: the first and foremost step of the safety analysis is collection of input data, formulation of technological process (TP) safety criteria and objectives. The selection of standards that will be applied to prove the safety level is justified in the frames of safety lifecycle planning.
2.  Preliminary safety analysis (PSA): All functions of technological process equipment are assessed to discover potential functional failures, and hazards connected with particular failure states are classified. The preliminary safety analysis systematizes requirements and criteria laid down in the contract (tender documentation) and provides a preliminary proof of that the proposed technological process equipment architecture can ensure fulfillment of these requirements, justifies the necessity of introducing protective measures, additional assemblies and functionality. The PSA is updated throughout the entire duration of the development process.
3.  Technological process safety analysis: collection, analysis and documenting the results, proving that the design, control system architecture, and selected components meet the safety requirements and objectives.
4.  Common cause analysis sets requirements for physical and functional separation, isolation, and independence of technological process elements.

The relation of the design process and the safety analysis is shown in Figure 1.

### 2.2. Risks Classification and Safety Barriers Design

Preliminary risk analysis is based on an assessment of potential hazards. Potential risks (the risk means a hazard containing a quantitative assessment of the frequency and severity of consequences) can be divided into different groups which are related to operational and technical (functional) hazards. The general list of risks necessary for the analysis is provided in ISO 12100 [5]. The risks can be considered as hazards associated with the equipment itself, with its failures and external hazards associated with the actions of operators, and the loss of electricity and power supply.

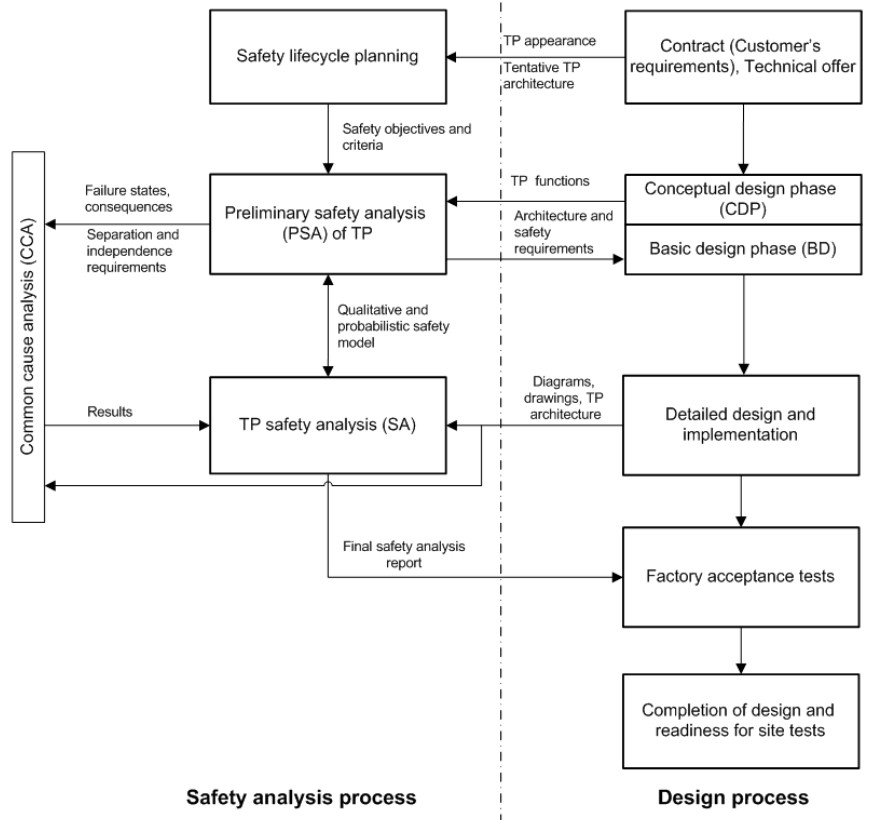

**Figure 1.** The relation of the design process and the safety analysis.

Reducing the risk and achieving the necessary level of safety is achieved by using a system of safety barriers. A recommended way to classify barrier systems is shown in Figure 2. However, note that active barrier systems are often based on a combination of technical and human/operational elements. Even though different words are applied, the classification in the fourth level in Figure 2 is similar to the classification suggested by Hale [6]. A safety barrier is a physical and/or non-physical means planned to prevent, control, or mitigate undesired events or accidents As regards the continuous time aspect, some barrier systems are available (functioning continuously), while some are off-line (need to be activated). Further, some barriers are permanent, while some are temporary. Permanent barriers are implemented as an integrated part of the whole operational life cycle, while temporary barriers only are used in a specified time period, often during specific activities or conditions.

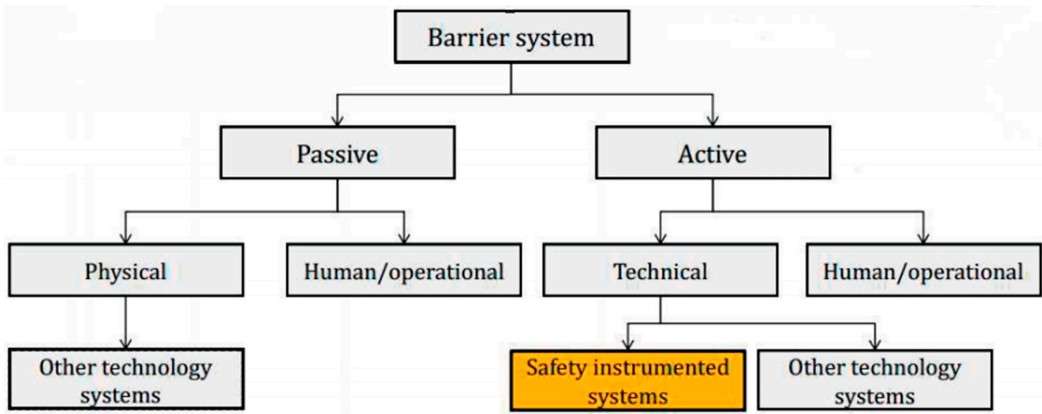

**Figure 2.** Safety barrier classification, adopted from [7].

Authors [8] note that identifying technical (physical) safety barriers, usually, is quite simple, but in the case where the safety barrier includes an action, for example, the operator's response to an alarm, you should be careful and distinguish between the action itself, which performs the barrier function, and the factors that help the operator in making the correct decision (technological instructions, training, precise information presentation, etc.) [9] offers a somewhat different approach classification of safety barriers based on evaluating their effectiveness in the event of a potentially dangerous situation. The degree of efficiency (high, medium, low) distinguishes the following types of safety barriers. Technical (high efficiency) barriers can prevent the spread of risk factors, reduce the risk of a situation, mitigate the consequences, or reduce the likelihood of risk factors [9]. Various technical barriers provide selective action against possible failures and external threats. The same applies to further escalation from the triggering event to consequences. The following subcategories are distinguished technical barriers: technical barriers that are triggered on demand (emergency cut-off valve, drencher system, emergency tank); technical passive, operate on a permanent basis, perform barrier function by its mere presence (safety valve, collapse, fire-proof and explosion-proof partitions etc.); technical control barriers that activate other barriers that prevent or mitigate the consequences of a dangerous event (gas detectors, fire alarm system, accident notification system, etc.).

Figure 3 shows how to develop requirements for safety barriers.

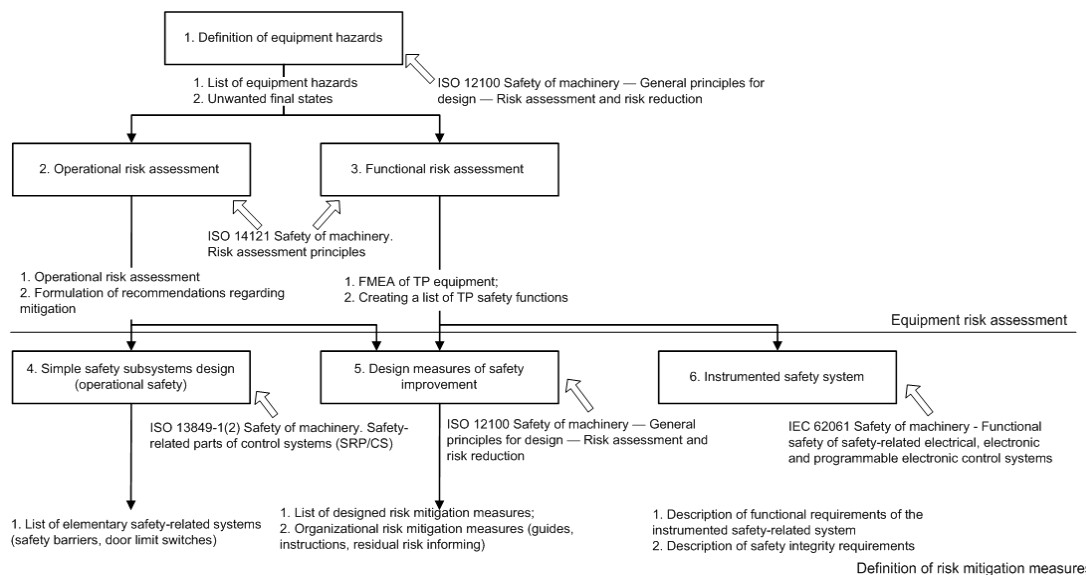

**Figure 3.** Procedure for the development of safety barriers.

The process of designing safety barriers takes place using [5,10–12].

Barriers of this type cannot prevent the development of the accident, but can activate other barriers that will do this. Human (organizational) (average efficiency) barriers contribute to the control of a process or activity. This type of barrier can reduce the probability of the triggering event by strengthening other barriers or preventing them from being weakened, but if a potentially dangerous event has already been initiated, then this type of barrier can prevent its development or reduce the consequences. The following subcategories are distinguished: types of barriers: procedural (inspections and observations, control tools, process management, work risk assessment, work permit system etc.); human (operational) (control by the operator, supervision, periodic detours, etc.); and fundamental (low efficiency in the immediate vicinity of the event). Their effect is divided in time from the occurrence of the threat to the implementation of the factor risk.

*2.3. Risk Reduction and SIS*

Risk reduction of Equipment under control (EUC) or technological process is shown in Figure 4.

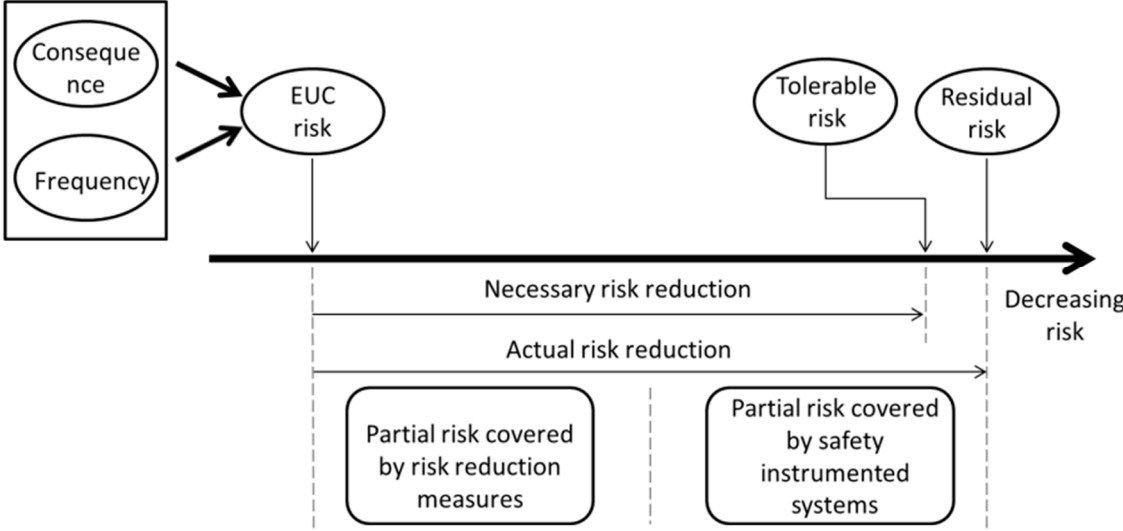

**Figure 4.** Risk reduction of equipment under control (EUC) or technological process.

However, fundamental barriers make a significant difference and an important and effective contribution to the safety of the system by providing checks and controls for vulnerable systems and the original causes of failures. The following subcategories are distinguished by these types of barriers: fundamental procedural (analysis of the project, assessment of commissioning, checking the internal regulations, analysis of operation, confirmation of qualification); and fundamental human (good health of workers, etc.) [11]. A number of standards and guidelines have been issued to assist in designing, implementing, and maintaining reliable SISs. The most important of these is the international standard [2], which is a generic standard that outlines key requirements to all phases of the SIS life-cycle. The approach to developing safety functions related to a computer instrumental safety system is shown in Figure 5.

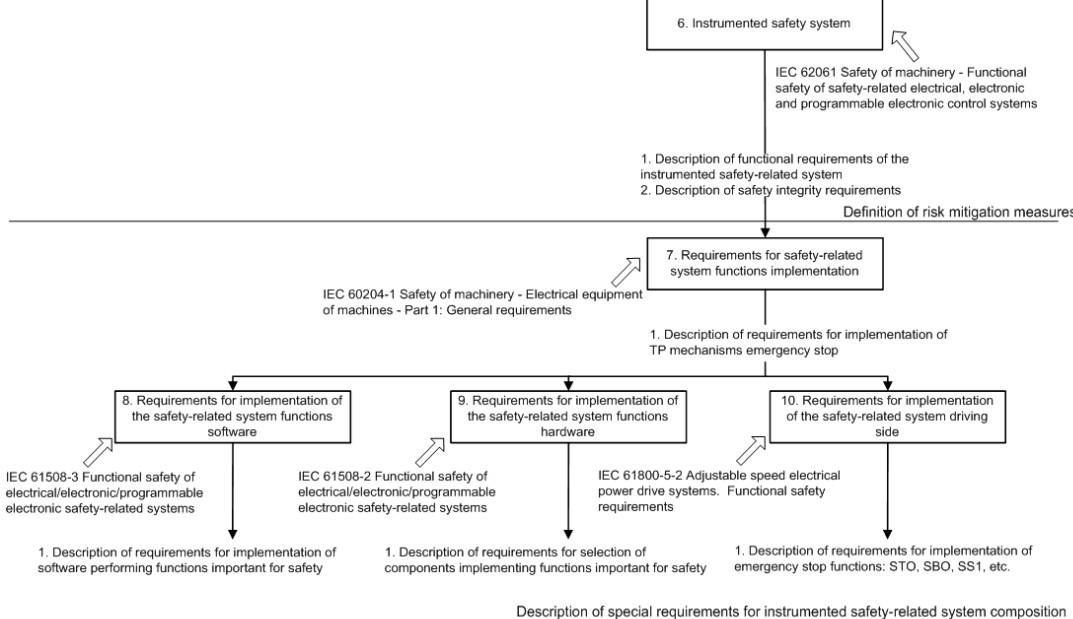

**Figure 5.** Risk reduction of technological process.

For some specific computer systems such as cluster computing systems, especially real-time, the key is to ensure reliability and fault tolerance while maintaining the continuity of the computing

process. The achievement of high and stable performance indicators, reliability, fault tolerance [13] and security of computer systems is facilitated by the use of technologies for consolidation of clustering and virtualization resources [14], accompanied by replication and migration of virtual machines between physical servers. Migration and replication of virtual machines speeds up the reconfiguration process after failures of physical resources and contributes to supporting the continuity of the computing process required for managing cyber-physical systems and real-time technological processes.

## 3. Risk Reduction and Optimization

### 3.1. Problem Statement

The problem of optimizing the composition of the safety barriers and SIS is to select the necessary and sufficient set of sensors, logic elements and final performers, taking into account the constraints on the budget of the project. It is considered that any safety measures, applying the principle of risk reduction down to acceptable level [15]. Which protective measure has an estimated level of risk reduction factor (RRF). The main objective of all protective measures is to provide protection and reduce the initial risk level to an acceptable level.

The level of risk reduction taking into account safety barriers is shown in the Figure 6.

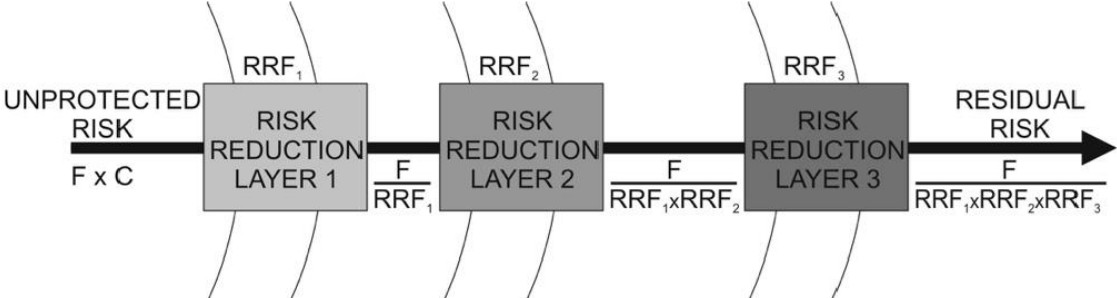

**Figure 6.** Model of risk reduction layers.

The purpose of this work is to solve the problem of optimization of the choice of a set of safety measures used in SIS, with the provision of specified safety requirements and cost.

The event tree (ETA) serves as a convenient and visual tool for representing security measures oriented to source events. This type of analysis has been widely used in probabilistic risk assessment of nuclear power plants. The application of ETA is described in detail in [16].

The known methods of HAZOP and LOPA are presented in the manuals [1,2] and works [16,17].

The probability of failure of safety measures can be determined by $q(t) = e^{-\lambda t}$, where $\lambda$ is the equipment failure rate. Cascading failures and common case failures are not considered in this approach.

In general, we can introduce:

$$
\begin{cases}
\min \left( \sum\limits_{i=1}^{n} S_i b_i \right) \\
\sum\limits_{i=1}^{n} (q_i) \cdot \left( \prod q_{lock_j}^{b_j} \cdot \prod q_{diag_j}^{b_j} \cdot \prod q_{ems_j}^{b_j} \right) < q_{req_1} \\
\cdots \\
\sum\limits_{i=1}^{n} (q_i) \cdot \left( \prod q_{lock_j}^{b_j} \cdot \prod q_{diag_j}^{b_j} \cdot \prod q_{ems_j}^{b_j} \right) < q_{req_n},
\end{cases}
\tag{1}
$$

$q_i$—probability of failure of the *i*-th component of the process system;

$S_j$—the cost of implementing the *j*-th safety measure;

$q_{lockj}$—the probability of failure of the *j*-th lock;

$q_{emsj}$—the probability of failure of $j$-th emergency stop;

$q_{diagj}$—probability of failure of the $j$-th diagnosis, revealing pre-emergency conditions; and

$q_{req}$—the probability of occurrence of a dangerous situation, specified in regulations or determined during the analysis.

### 3.2. Approach to the Optimization Problem Solving

The problem of optimization of the choice of safety measures is a modification of the "Backpack Problem" [18], a class of combinatorial optimization problems, which can be formulated as follows:

$$\max_{x} \sum_{j=1}^{n} p_j x_j, \quad x_j \in \{0,1\}, \quad j = 1, \ldots, n$$
$$\sum_{j=1}^{n} \omega_{i,j} x_j \leq c_i, \quad i = 1, \ldots, m \tag{2}$$

where $p_j$ and $\omega_{i,j}$ are weights, and $c_i$ is a cost, and $x = (x_1, \ldots, x_n)$.

The backpack problem can be solved in several ways: the method of dynamic programming [19]; brute force; the method of branches and boundaries [20]; the method of statistical modeling. Consider the application of the statistical modeling method. In general, the approach can be represented as follows, find the maximum of the function $S(x)$ on a given set $X$. Let us assume that the maximum is achieved for only one value of the parameter x*. Let us denote the maximum by $\gamma^*$.

$$S(x^*) = \gamma^* = \max_{x \in X} S(x) \tag{3}$$

Optimization problem can be related to the calculation of probability l = P(S(X) ≥ γ), where X has some probability density f(x; u) on the set X (for example, having a uniform distribution density) and $\gamma$ is close to the unknown $\gamma^*$. As is correct, l is the probability of a rare event, so a sampling-by-significance approach can be used. Thus, sampling from such a distribution yields optimal or nearly optimal values. The last value $\gamma^* = \gamma$ is usually unknown, but using statistical modeling, a sequence $\hat{\gamma}_t$ is formed at each step of the simulation, which tends to the optimal $\gamma^*$, as well as at each step the change of the modeled vector $\hat{v}^*$ is fixed [21–23].

### 3.3. Algorithm of Monte Carlo Simulation

1. Choose the initial vector of parameters $\hat{v}_0$, let elite selection be $N^e = \lceil \varrho N \rceil$, $\varrho$-parameter. Take the counter $t = 1$;

2. Generate $N$ random vectors $X_1, \ldots, X_N$ with density $f(\cdot; \hat{v}_{t-1})$, determine the values of effect $S(X_i)$ for all i, and arrange them in ascending order from smaller to larger: $S_{(1)} \leq \cdots \leq S_{(N)}$.

Let $\gamma_t$ be the (1 − e) quintile of the obtained values, thus $\hat{\gamma}_t = S_{(N-N^e+1)}$;

3. Using the same sample of random vectors $X_1, \ldots, X_N$ solve the equation $\max_{v} \frac{1}{N} \sum_{k=1}^{N} I_{\{S(X_k) \geq \hat{\gamma}_t\}}$ $\ln f(X_k; v)$ denote the solution as $\hat{v}_t$, where I is indicator function (I = 1 if $S(X_k) \geq \hat{\gamma}_t$, and 0 otherwise)

4. If the stop criterion is reached, then end the algorithm, otherwise change the counter $t = t + 1$ and proceed to step 2.

## 4. Model of Technological Process Subsystem

### 4.1. Model Description

As an example, we will consider the fuel supply subsystem, which includes a fixed volume tank (Tank), a level sensor (LV), a pumping valve to the next section of the process (V1), and a feed pump (PD) with a control system implemented on the control unit (CU). During the preliminary analysis, it was revealed that two dangerous conditions are possible at this site: the occurrence of a fire and its propagation, as well as tank overflow. Assume that the required probability of

preventing the development of fire and exceeding the level in the tank should be less than $1 \times 10^{-5}$ and $1 \times 10^{-4}$ per year, respectively. Modeling of safety-related systems is based on the theory of reliability. To describe the possible consequences of failures of the main equipment, FMEA analysis of the subsystem equipment is used, the analysis is performed for the operating mode. The qualitative analysis as the failure mode and effect analysis (FMEA) of the technological process subsystem in accordance with [1] is given in Table 1.

**Table 1.** FMEA of technological subsystem.

| Element | Failure Type | Consequences | Safety Measures |
|---|---|---|---|
| Tank | Destruction of the hull | Fire | D1-control of the hull by ultrasonic control device<br>D2-magneto resistive monitoring device<br>H1-switching on the fire pump and water supply<br>H3-emergency opening of the emergency drain |
| Level sensor | False values | Exceeding the limit | D5-monitoring of the sensor<br>Z2-emergency stop of process equipment (pump)<br>H3-emergency opening of drain valve |
| Level sensor | The absence of values | Shutdown | not required |
| Feed pump | Feed loss | Shutdown | not required |
| Feed pump | Overheat | Fire | D3-monitoring the state of the windings<br>D4-housing temperature control<br>H1-switching on the fire pump and water supply |
| Feed pump | False start | Exceeding the limit | Z2-emergency stop of process equipment (pump)<br>H3-emergency opening of drain valve |
| Transfer valve | Failure to respond | Shutdown | not required |
| Transfer valve | False opening | Shutdown | not required |
| Control system | Loss of control signal | Shutdown | not required |
| Control system | Erroneous command | Exceeding the limit | Z2-emergency stop of process equipment (pump)<br>L1-pump control limitation when 70% of the tank volume<br>H3-emergency opening of drain valve |

Safety measures D1–D3 to ensure control are taken continuously.

Following methods for assessing reliability: Quantitative evaluation using simplified equations based on block diagrams of reliability and analysis of failure trees [24]. In some cases, Markov analysis can be used, a more complex approach allows working with dynamic models that take into account the development of failure over time [25]. Taking into account various variants of implementation of safety measures it is possible to receive the following optimization problem [26]:

$$
\begin{cases}
\min \left( \sum\limits_{j=1}^{9} s_j b_j \right) \\
\left( q_{tank} \right) \cdot q_{D_1}^{b_1} q_{D_2}^{b_2} q_{Z_1}^{b_6} q_{Z_3}^{b_8} + \left( q_{PD.H} \right) \cdot q_{D_3}^{b_3} q_{D_4}^{b_4} q_{Z_1}^{b_6} < q_{fire} = 1 \cdot 10^{-5} \\
\left( q_{LV.F} \right) \cdot q_{D_5}^{b_5} q_{D_2}^{b_2} q_{Z_3}^{b_8} + \left( q_{PD.F} \right) \cdot q_{Z_2}^{b_7} q_{Z_3}^{b_8} + \left( q_{CU.F} \right) \cdot q_{Z_2}^{b_7} q_{Z_3}^{b_8} q_{L_1}^{b_9} < q_{o.l.} = 1 \cdot 10^{-4}
\end{cases}
\tag{4}
$$

It is needed to find the vector B = $\{b_1, b_2 \ldots b_9\}$, at which (1) is executed, on a set of initial data from Tables 2 and 3. For example, the vector B = $\{1, 0, 1, 0, 0, 0, 1, 0, 0\}$ means that, as part of the

safety instrument system, safety measures are used: monitoring the condition of the tank body by the ultrasonic method (D1), monitoring the condition of the feed pump windings (D3), emergency opening of the drain valve (Z3). The total number of combinations $2^9 = 512$. In this example, for clarity, the number of options is not so large; in real systems, the number of combinations can reach very large values.

### 4.2. Model Initial Data

The initial data on the reliability of the equipment of the production line and safety measures are presented in Tables 2 and 3, respectively.

The fuel supply subsystem works 8760 h a year, without safety measures: $q_{fire} = 4.36 \times 10^{-2}$, $q_{o.l.} = 7.43 \times 10^{-3}$.

To assess the effect of safety measures, risk reduction indicators are used, expressed in the probability of failure of the safety barrier per year. To conclude on the achieved level of security completeness, it is necessary to additionally consider the safety architecture and the level of diagnostic coverage.

**Table 2.** Dangerous failure rate

| Event | Code | FR, h$^{-1}$ | $\alpha$ * | Probability Per Year ** |
|:---:|:---:|:---:|:---:|:---:|
| Tank. Destruction | $q_{tank}$ | $1 \times 10^{-7}$ | 80% | $7.01 \times 10^{-4}$ |
| Feed pump. Overheating | $q_{PD.H}$ | $1 \times 10^{-5}$ | 50% | $4.29 \times 10^{-2}$ |
| Level sensor. False signal | $q_{LV.F}$ | $1 \times 10^{-6}$ | 30% | $2.62 \times 10^{-3}$ |
| Feed pump. False start | $q_{PD.F}$ | $1 \times 10^{-5}$ | 5% | $4.37 \times 10^{-3}$ |
| Control system. Erroneous response | $q_{CU.F}$ | $1 \times 10^{-6}$ | 5% | $4.38 \times 10^{-4}$ |

* The rejection rate was accepted in accordance with FMD-2013, ** The reliability of measures is based on the typical values of reliability of equipment intended for such tasks. The NPRD-2016 database and data on the reliability of the main manufacturers of electrical products were used as initial data.

**Table 3.** Baseline data on safety measures

| # | Safety Measures | Cost, c.u. | Probability Per Year * |
|:---:|:---:|:---:|:---:|
| q$_{D1}$ | Control of the body condition by ultrasonic method | 100 | $1.00 \times 10^{-3}$ |
| q$_{D2}$ | Magneto resistive monitoring device | 200 | $1.00 \times 10^{-3}$ |
| q$_{D3}$ | Control condition of winding | 10 | $1.00 \times 10^{-5}$ |
| q$_{D4}$ | Housing temperature control | 25 | $1.00 \times 10^{-4}$ |
| q$_{D5}$ | Monitoring of the sensor status by initial test | 10 | $1.00 \times 10^{-5}$ |
| q$_{Z1}$ | The inclusion of the fire pump and water flow | 400 | $1.00 \times 10^{-3}$ |
| q$_{Z2}$ | Emergency stop of process equipment (pump) | 200 | $1.00 \times 10^{-3}$ |
| q$_{Z3}$ | Emergency opening of the discharge valve | 200 | $1.00 \times 10^{-4}$ |
| q$_{L1}$ | Pump control limitation at 70% of tank volume | 5 | $1.00 \times 10^{-4}$ |

* The reliability of measures is based on the typical values of reliability of equipment intended for such tasks. The NPRD -2016 database and data on the reliability of the main manufacturers of electrical products were used as initial data.

### 4.3. Optimization Parameters

For optimization we introduce a single target function:

$$S(x)\ \beta \sum_{i=1}^{m} I_{\{\sum_j \omega_{i,j} x_j \geq c_i\}} + \sum_{j=1}^{n} p_j x_j, \tag{5}$$

where $\beta = -\sum_{j=1}^{m} p_j$. In this case, $S(x) < 0$ if one of the inequalities fails and $S(x) = \sum_{j=1}^{n} p_j x_j$, if satisfied. Since the vector x is binary, the multivariate Bernoulli distribution with density $f(x,v) = \prod_{j=1}^{n} v_j{}^{x_j}(1-v_j)^{1-x_j}$ is chosen as the initial distribution. As initial parameters we will accept the following $N = 10^2$ and $N^e = 10$, and $\hat{v}_0 = (1/2, \ldots, 1/2)$.

We will not use the mixing parameter to define $\hat{v}_t$ ($\alpha = 1$), so at each iteration $\hat{v}_t$ will be as follows:

$$\hat{v}_{t,j} = \frac{\sum_{k=1}^{N} I_{\{\hat{S}(X_k) \geq \hat{\gamma}_t\}} X_{k,j}}{\sum_{k=1}^{N} I_{\{\hat{S}(X_k) \geq \hat{\gamma}_t\}}}, \ j = 1, \ldots, n \tag{6}$$

where $X_{k,j}$ is the *j*-th component of the *k*-th random vector X. The expression is used as a stop criterion $d_t = \max_{1 \leq j \leq n} \{\min\{\hat{v}_{t,j}, \ 1 - \hat{v}_{t,j}\}\} \leq 0.01$. For each population t of generated values, we calculate the threshold $\hat{\gamma}_t$, the largest value $S(X_k)$, and the value of the stop criterion $d_t$.

### 4.4. Modeling Results

To demonstrate the convergence of the method, independent modeling iterations were performed. In each cycle, changes in the density of the vector $\hat{v}_t$ were recorded after calculation using Equation (6). Figure 7 present the average change value of the parameter vector while 100 independent iteration. The final decision, the value of the vector $\hat{v}_t$ corresponds to the following composition of equipment and measures: the application of monitoring the condition of the pump windings, and the emergency opening of the drain valve. Vector B = {0, 0, 1, 0, 0, 0, 1, 0, 0} is optimal, with a total cost of S = 210, and $q_{fire} = 4.99 \times 10^{-7}$ and $q_{o.l.} = 7.43 \times 10^{-7}$. The results of the dynamics of the vector $\hat{v}_t$ during updating after each modeling cycle of 100 iterations is presented in Figure 8.

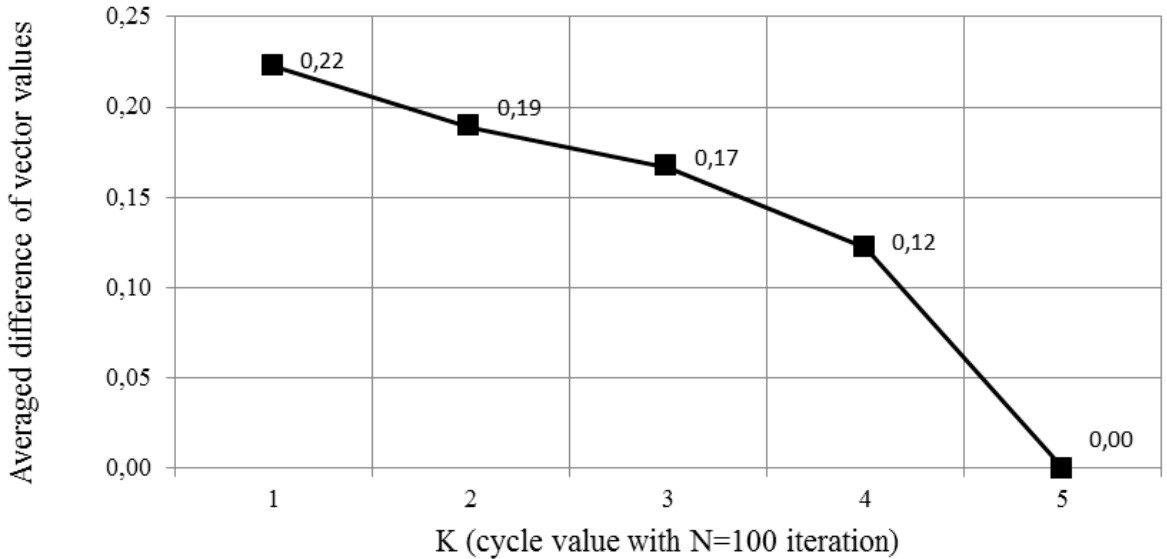

**Figure 7.** Averaged difference of vector values.

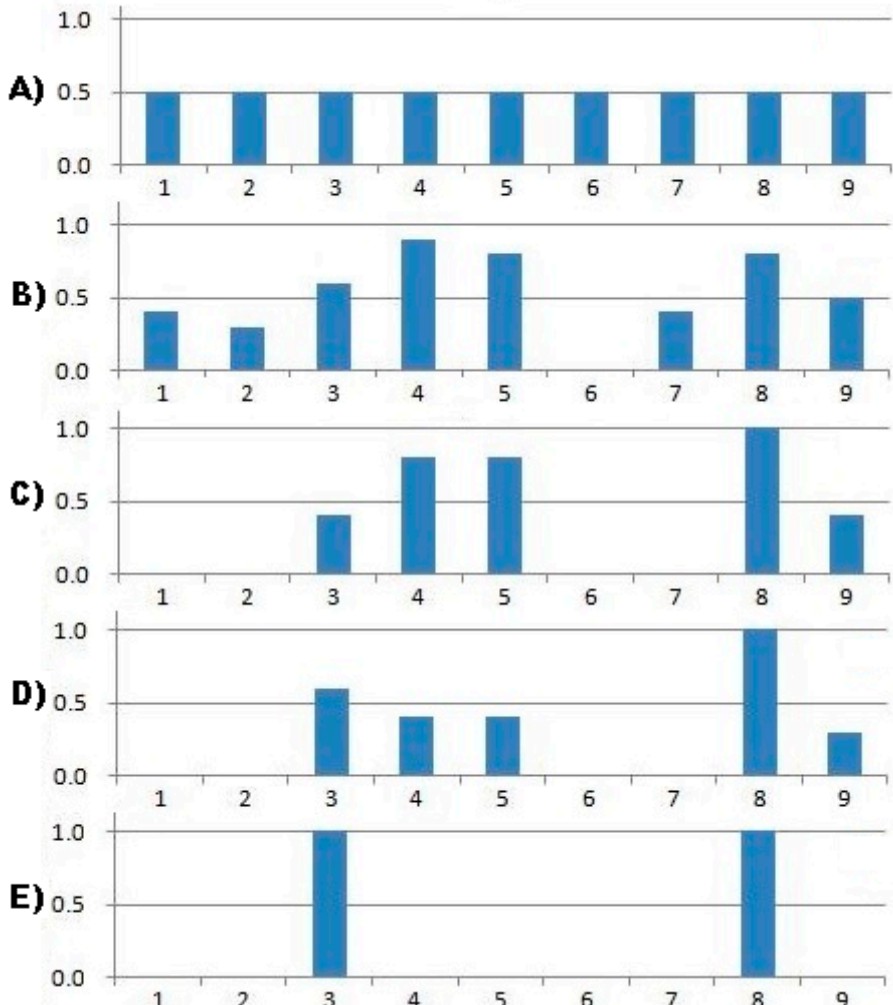

**Figure 8.** Dynamics of the probability vector $\hat{v}_t$: (**A**) initial state, (**B**) modified after 1st cycle, (**C**) modified after the 2nd cycle, (**D**) modified after the 3rd cycle, (**E**) final state.

## 5. Conclusions

This paper presents a method of bringing the problem of the optimization of a set of safety measures provided in the SIS to the problem of discrete optimization. The method of statistical modeling with significance sampling was used as a solution method. The obtained solution corresponds to the solution obtained by brute force. The obtained result can serve as a basis for the development of the requirements specification in accordance with the requirements for the life cycle of the system. Development of a risk model, including safety barriers that may prevent, control, or mitigate accident scenarios with in-depth modeling of the barrier performance allows explicit modeling of functional common cause failures (e.g., failures due to functional dependencies on a support system). The classification of safety measures is given, and the model of risk reduction based on deterministic analysis of the process is considered. It is shown that the task of changing the composition of safety measures can be represented as the knapsack discrete optimization problem, and the solution is based on the Monte Carlo method. A numerical example is provided to illustrate the approach. The considered example contains a description of failure conditions, an analysis of the types and consequences of failures that could lead to accidents, and a list of safety measures. Solving the optimization problem used real reliability parameters and cost of equipment. Based on the simulation results, the optimal composition of safety measures providing cost minimization is given. For the future research, the authors plan to take into

account the dynamic change of the system, e.g., under cyberattacks which aim to compromise the safety features of the system.

**Author Contributions:** Conceptualization: V.B.; methodology: A.M.; writing—original draft preparation: A.M.; writing—review and editing: A.M.; visualization: A.M. All authors have read and agreed to the published version of the manuscript.

**Funding:** This research received no external funding.

**Conflicts of Interest:** The authors declare no conflict of interest.

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
