# Peer review of "Risk Reduction Optimization of Process Systems under Cost Constraint Applying Instrumented Safety Measures†"

_computers, doi:10.3390/computers9020050_

Round 1

Reviewer 1 Report

The paper presents a calculation method to optimize risk reduction of a process system applying instrumented safety methods under costs constraint. The title of the paper is not specific enough. For example it could be modified to: "Risk reduction optimization of process systems under cost constraint applying instrumented safety measures". It is not fully clear what hazards authors are addressing. Initially, they mention process safety but later they write techological processes. So, authors have to make clear at the start what is the scope of the paper. The focus of the paper seems initally on ESD and SISs. Please give some examples of SISs, because in the process industry it usually is one SIS or two SIS's at specified SIL levels in an array of other barriers and residual risk is calculated for the whole array. Therefore it is recommended to describe the various options of risk reduction first and zoom in on instrumented safety measures later. In fact, as the article is now, only at the end when you describe the case, it becomes clear what you mean. Figure 2 shows the various types of barriers, alright, but what is lacking is a protection layers event tree. Figure 6. The DCS process control is to keep the process in optimum conditions and within safety margins, but is itself not part of the protective safety layers. These are separate and should be fully independent. The right hand side of Fig 6B is 'floating in the air'. What do you mean? After an ESD the process has stopped. In principle, you apply the known layer of protection analysis (LOPA) after having identified hazard events, which can be achieved with FMEA and HAZOP techniques. Why don't you refer to the large volume of literature on these topics. Figure 6A shows a typical LOPA configuration. However, you must mention the condition that this only holds when the individual failures of the measures are independent of each other. Section 3.3 is too short and not all symbols are explained. More should be said on why and how, although you have applied a known method you cannot assume that all your readers are familiar with this. Do you assume that measures D1 and other D's in Table 1 are continuously monitoring? Please comment on the reliabilities of the safety measures in comparison with the ones of the equipment to be protected? Some are of the same order of magnitude. What SIL level do the SISs have? Why do you mention the optimization method Cross entropy Monte-Carlo only in the Conclusions and not on line 170? The symbol I is not explained in equations 5 and 6.
Minor comments:
Line 36: Is it 'finite' or 'final' element controls?
Line 73: can 'be' divided
Lines 96-98: The sentence is in bad English.
Line 149: ALARP is specifically English. It is preferable to use instead the expression "down to acceptable level".
Line 176: Is the rare event assumed not to occur, so it is residual risk?

Author Response

Dear Mr. Reviewer

Reviewer 2 Report

1) One of the FMEA-family analysis methods' limitations is that its results are valid at a certain time of lifecycle. And so it is crucial to point which stage of lifecycle (in the simplest way e.g. - development, operation, maintenance, commissioning, etc.) are covered by the proposed solution. 

2) How to apply the proposed solution in dynamics? E.g. if the event occurred and the countermeasures will be implemented the developer needs to justify that the system is adequately safe in a given condition in a certain environment.

3) Fig.7 should be split into 5 parts and should be named accordingly (e.g. a,b,c,d,e)

4) The title "Risk Reduction Approach to Design Computer Based
Safety Systems" is a little bit wider than the content. I'd recommend reviewing it once more and shift from "Design" concept to "Evaluation" concept because the paper is not much about designing safety systems, but about the safety evaluation of safety-related systems. 

So, for the future research, I'd propose to take into account the dynamic change of the system (you can even talk about resilience), e.g. under cyberattacks, which are aiming to compromise the safety features of the system. 

Author Response

Dear Mr. reviewer

Thank you very much for your friendly attitude to our article and for your valuable comments, which are undoubtedly aimed at improving the presentation of the results of the article.

Round 2

Reviewer 1 Report

Previous comments have been adequately dealt with. Thank you. It made the paper more reader-friendly.

At the bottom part of the Conclusion, please correct the typo 'autors' into 'authors'.